# The Role of Hyperthermic Intraperitoneal Chemotherapy in Uterine Cancer Therapy

**DOI:** 10.3390/ijms241512353

**Published:** 2023-08-02

**Authors:** Iason Psilopatis, Christos Damaskos, Nikolaos Garmpis, Kleio Vrettou, Anna Garmpi, Panagiotis Sarantis, Evangelos Koustas, Efstathios A. Antoniou, Gregory Kouraklis, Athanasios Chionis, Konstantinos Kontzoglou, Dimitrios Dimitroulis

**Affiliations:** 1Department of Obstetrics and Gynecology, University Erlangen, Universitaetsstrasse 21–23, 91054 Erlangen, Germany; 2Second Department of Propedeutic Surgery, Laiko General Hospital, Medical School, National and Kapodistrian University of Athens, 11527 Athens, Greece; 3Nikolaos Christeas Laboratory of Experimental Surgery and Surgical Research, Medical School, National and Kapodistrian University of Athens, 11527 Athens, Greece; 4Renal Transplantation Unit, Laiko General Hospital, 11527 Athens, Greece; 5Department of Cytopathology, Sismanogleio General Hospital, 15126 Athens, Greece; 6First Department of Propedeutic Internal Medicine, Laiko General Hospital, Medical School, National and Kapodistrian University of Athens, 11527 Athens, Greece; 7Molecular Oncology Unit, Department of Biological Chemistry, Medical School, National and Kapodistrian University of Athens, 11527 Athens, Greece; 8Department of Surgery, Evgenideio Hospital, Medical School, National and Kapodistrian University of Athens, 11527 Athens, Greece; 9Second Department of Gynecology, Laiko General Hospital, Medical School, National and Kapodistrian University of Athens, 11527 Athens, Greece

**Keywords:** uterine, endometrial, cancer, sarcoma, hyperthermic, intraperitoneal, chemotherapy

## Abstract

Endometrial cancer and uterine sarcoma represent the two major types of uterine cancer. In advanced stages, both cancer entities are challenging to treat and correlate with a meagre survival and prognosis. Hyperthermic Intraperitoneal Chemotherapy (HIPEC) is a form of localized chemotherapy that is heated to improve the chemotherapeutic effect on peritoneal metastases. The aim of the current review is to study the role of HIPEC in the treatment of uterine cancer. A literature review was conducted using the MEDLINE and LIVIVO databases with a view to identifying relevant studies. By employing the search terms “hyperthermic intraperitoneal chemotherapy”, “uterine cancer”, “endometrial cancer”, and/or “uterine sarcoma”, we managed to identify 26 studies published between 2004 and 2023. The present work embodies the most up-to-date, comprehensive review of the literature centering on the particular role of HIPEC as treatment modality for peritoneally metastasized uterine cancer. Patients treated with cytoreductive surgery, alongside HIPEC, seem to profit from not only higher survival but also lower recurrence rates. Factors such as the completeness of cytoreductive surgery, the peritoneal cancer index, the histologic subtype, or the applied chemotherapeutic agent, all influence HIPEC therapy effectiveness. In summary, HIPEC seems to represent a promising treatment alternative for aggressive uterine cancer.

## 1. Introduction

Uterine cancer includes two types of cancer: endometrial cancer and uterine sarcoma.

### 1.1. Endometrial Cancer

The most frequent malignant female genital tract tumor in the US is endometrial cancer. The American Cancer Society predicts that in the United States in 2023 there will be approximately 66,200 new cases of uterine body cancer diagnosed and 13,030 women will pass away from uterine cancer [1]. The majority of postmenopausal women with endometrial cancer are 55 to 64 years old, with 63 years old being the median age at diagnosis [2]. Type I endometrial carcinoma resulting from atypical endometrial hyperplasia and type II endometrial cancer of non-endometrioid histology are the two distinct histopathologic categories into which endometrial cancer can be subdivided [3]. In addition to being correlated with microsatellite instability, Kirsten Rat Sarcoma Virus (KRAS), -catenin, and/or human MutL Homolog 1 (hMLH1)/MutS Homolog 2 (MSH2) mutations, type I endometrial carcinoma is directly linked to prolonged exposure to high estrogen levels. The majority of type II endometrial cancers are estrogen-independent, develop from atrophic endometrium in postmenopausal patients, and are linked to p53 mutations, p16, and E-cadherin inactivation, as well as HER2 amplification [4,5]. While adjuvant chemotherapy must be given to patients with high-intermediate or high-risk endometrial cancer, as well as advanced and/or recurrent disease, surgery is advised as a monotherapy for low-risk endometrial malignancies [6]. The first-line chemotherapy protocol calls for a combination of doxorubicin, cyclophosphamide, or cisplatin, followed by carboplatin and paclitaxel [7].

Despite the high reported rates of chemotherapy response, this response only lasts between 4 and 8 months on average [8,9,10,11], while the American Cancer Society estimates that the 5-year overall survival rate is 84% [12]. However, with 5-year survival rates falling to 20% in cases of distant metastasis, the prognosis for women with advanced disease remains dire [12].

### 1.2. Uterine Sarcoma

High-grade malignant tumors called uterine sarcomas develop from the uterus’ smooth muscles and/or connective tissue [13]. Uterine sarcomas can be divided into four different groups depending on the type of cells they originate from. The most prevalent type of uterine leiomyosarcomas originates from the myometrium. According to the characteristics of the cancer cells and the growth pattern of the tumor, endometrial stromal sarcomas can be divided into low- and high-grade tumors after developing in the uterus’ endometrial stroma. Undifferentiated sarcomas typically grow and spread quickly, and they can develop from either the endometrium or the myometrium. Adenosarcomas are biphasic neoplasms made up of benign epithelial components and a malignant, typically low-grade mesenchymal component, though high-grade sarcomatous overgrowth can occur [13].

Leiomyosarcomas and endometrial stromal sarcomas are the most prevalent types of uterine sarcomas, which make up 2–5% of all uterine cancers [14]. Afro-American women are twice as likely as Caucasian women to develop uterine leiomyosarcomas [14]. An increased risk of uterine sarcomas has been associated with prior pelvic radiation therapy, tamoxifen use, congenital retinoblastoma, hereditary leiomyomatosis, and renal cell cancer syndrome [15]. Numerous uterine sarcomas have so far been associated with particular chromosomal translocations, and the fusion genes that result from these translocations activate crucial transcription factors [16].

The non-specific signs and symptoms of uterine sarcomas can be mainly attributed to uterine changes that are not cancerous, endometrial hyperplasia, or endometrial cancer. The most frequent symptom is abnormal bleeding or spotting, especially after menopause, followed by vaginal discharge, pain, feeling of a mass, and urinary or bowel issues [17].

A transvaginal ultrasound, magnetic resonance imaging (MRI), and eventually a positron emission tomography (PET) scan are used in the diagnostic evaluation of uterine sarcomas in addition to a physical examination. However, in order to determine the tumor grade and the hormone receptor status, hysteroscopic endometrial biopsy and tissue sampling are always necessary for a definitive diagnosis [18]. High serum levels of the markers Cancer Antigen 125 (CA-125), Lactate Dehydrogenase (LDH), C-reactive protein (CRP), and D-dimers could theoretically indicate uterine sarcoma but are strongly influenced by various other factors and, consequently, lack specificity, according to Liu et al. in their recent review of the literature focusing on advancements in the preoperative identification of uterine sarcoma [19]. Despite being a cheap and practical screening technique, ultrasound may not be able to definitively determine whether uterine masses are benign or malignant. On the other hand, MRI has excellent soft-tissue resolution, and some types of degenerative uterine fibroids have similar signal intensities, which contributes to a certain rate of misdiagnosis. Last but not least, PET-CT ensures the highest accuracy but is expensive and difficult to promote [19].

The mainstay of treatment for patients with early-stage resectable uterine leiomyosarcoma and undifferentiated sarcoma is hysterectomy combined with bilateral salpingo-oophorectomy. When a sarcoma recurrence is very likely, adjuvant radiochemotherapy may be used to finish the course of treatment. Systemic therapy is typically used to treat patients with advanced disease, particularly when complete surgical excision is not possible. When hormone receptor status is positive, hormonal therapy is added to the similar treatment approach for endometrial stromal sarcomas [20].

### 1.3. Hyperthermic Intraperitoneal Chemotherapy

Hyperthermic Intraperitoneal Chemotherapy (HIPEC) is a form of localized chemotherapy that is heated to improve the chemotherapy’s penetration and cytotoxicity against the tumor cells [21]. The goal of HIPEC is to completely clear the peritoneal surface of any microscopic disease that may still be present [21]. By enhancing the cytotoxicity of some chemotherapeutic agents and extending the depth of chemotherapy penetration into tumor nodules, moderate hyperthermia above 41 °C has a direct anti-tumor effect [22]. Temperature probes are inserted at various locations throughout the procedure, including the heat generator, the inflow and outflow drains, the bladder, the liver, and the mesentery [22]. Both an open and a closed abdomen can undergo HIPEC. In both perfusion models, cytostatic solution heated to 41 to 43 °C is continuously infused into the abdominal cavity through a drainage system made up of inlet and outlet catheters [22]. Concerning the open abdomen technique, a Tenckhoff catheter and four closed suction drains are inserted through the abdominal wall at the conclusion of surgical cytoreduction. For intraperitoneal temperature monitoring, the temperature probes are fastened to the skin’s edge. To keep the abdominal cavity open, the skin edges at the level of the incisions are suspended until the Thompson self-retaining retractor by a monofilament. A plastic sheet is put into this suture to stop the leakage of the chemotherapy solution [22]. During the closed abdomen technique, the skin edges of the laparotomy are tightly sutured to allow perfusion in a closed circuit, but thermal catheters and probes are still placed in the same manner. The surgeon manually shakes the abdominal wall during the infusion to distribute heat evenly. In this method of establishing the circuit, the perfusate volume is greater and higher abdominal pressure is obtained during the perfusion, which helps the drug penetrate the tissue. The abdomen is reopened following infusion in order to remove the perfusate and prepare for the anastomosis. As there is little heat loss, this method allows for the rapid attainment of hyperthermia to be maintained [22]. Intraperitoneal chemotherapy’s main objective is to get rid of these free tumor cells and any invisible micrometastases [23]. Traditionally, HIPEC is carried out following surgical resection and prior to gastrointestinal tract reconstruction [24]. The concept of multimodal therapy is completed by this potentially curative therapeutic option, which is becoming more and more important for patients with peritoneal metastatic gastrointestinal and gynecological tumors and primarily peritoneal malignancies. A good pharmacokinetic profile, no cell cycle specificity, and absence of local peritoneal toxicity are the ideal characteristics of chemotherapy drugs for HIPEC [22] (Figure 1). Of interest, HIPEC seems to change the tumor microenvironment, induce differential gene expression in metastatic tumors or even significantly upregulate immune-related pathways, and downregulate DNA repair and homologous replication pathways [25].

In the field of gynecologic oncology, HIPEC has, to date, been mainly employed in the treatment of ovarian cancer. Only in 2023, numerous study groups have already published (systematic) review articles and/or meta-analyses on the role of HIPEC in ovarian cancer and have highlighted the improved survival outcomes after treatment with HIPEC in selected patients [25,26,27]. Margioula-Siarkou et al. recently published their review article on the role of HIPEC for gynecological malignancies with a focus on primary/recurrent ovarian, endometrial and cervical cancer, as well as peritoneal sarcomatosis, but mainly emphasized ovarian cancer and only exemplarily presented original research works of uterine cancer [28].

### 1.4. Aim of the Review

HIPEC has, so far, been reported to exhibit promising therapeutic results in the treatment of various cancer entities. To our knowledge, no comprehensive review of the literature has, however, been published on the role of HIPEC in uterine cancer therapy. The present work represents the most inclusive, up-to-date literature review on the advantages of HIPEC for the treatment of uterine cancer.

## 2. Materials and Methods

The literature review was conducted using the MEDLINE and LIVIVO databases. Solely original research articles written in the English language that explicitly reported on the role of HIPEC in uterine cancer were included in the data analysis. Studies focusing purely on the role of HIPEC in cancer entities other than uterine cancer or which did not explicitly specify the treated cancer entities were excluded. By employing the search terms “hyperthermic intraperitoneal chemotherapy”, “uterine cancer”, “endometrial cancer”, and/or “uterine sarcoma”, we were able to identify a total of 113 (duplicate records removed) articles published between 1999 and 2023. After the abstract review, 49 records were discarded in the initial selection process. The full texts of the remaining 64 publications were assessed and a total of 26 relevant studies meeting the inclusion criteria and published between 2004 and 2023 were selected for the final literature review. Figure 2 schematically depicts the aforementioned selection process.

## 3. Results

### 3.1. The Role of HIPEC in Endometrial Cancer Therapy

A total of 14 articles reported on the role of HIPEC in endometrial cancer therapy.

Abu-Zaid et al. reported on six patients with peritoneal carcinomatosis arising from endometrial cancer who were managed with standard peritonectomy procedures and visceral resections, alongside HIPEC with cisplatin and doxorubicin. Only one of the six patients experienced a disease relapse with metastases to hepatic, pelvic, and mesenteric lymph nodes, and died five months later. Another woman developed hepatic metastases within three months but was still alive at a follow-up of six months. The remaining 4 patients were alive and disease-free without evidence of recurrence after 7–35 months [29]. Furthermore, Bakrin et al. [30] treated five patients with recurrent endometrial cancer with cisplatin and mitomycin C after complete cytoreductive surgery and noted that one patient with pseudosarcomatous component experienced recurrent disease ten months post-operatively and died two months later. Another patient developed early recurrence with a malignant pleural effusion and passed away, whereas 3 patients were alive and disease-free even 39 months after surgery with a relatively good performance status [30]. Moreover, Brind’Amour et al. presented a group of three patients who underwent cytoreductive surgery and carboplatin HIPEC as their initial treatments for endometrial cancer and synchronous peritoneal metastases. Each patient received a wholly successful cytoreductive procedure. At 12 and 18 months, 2 patients passed away, and at 29 months, 1 patient did not still show a disease relapse [31]. Additionally, Chambers et al. examined the efficacy and safety of HIPEC with cisplatin and/or paclitaxel after complete cytoreductive surgery in seven uterine serous carcinoma patients following neoadjuvant chemotherapy and reported a good treatment tolerance with a median progression-free survival of 14 months and a median overall survival of 27 months [32]. Cornali et al. surprisingly treated a total of 33 patients with peritoneal metastases from endometrial cancer with cytoreductive surgery plus cisplatin-based HIPEC and concluded that completeness of cytoreduction with a low peritoneal cancer index is the only significant independent factor determining overall survival. Notably, in the follow-up period of 73 months, a 5-year overall survival of 30% and progression-free survival of 15.5% were documented for the included study population [33]. Delotte et al. [34] administered HIPEC to 13 endometrial cancer patients after cytoreductive surgery. One patient out of the thirteen who received treatment was lost to follow-up. A total of 3 patients passed away within the first year of treatment, and 2 patients passed away 12 and 19 months after their HIPEC procedure, respectively. Between 1 and 125 months after surgery, 7 patients were still alive, 4 of whom had not experienced a recurrence. For peritoneal carcinomatosis of endometrial origin, the peritoneal cancer index and the completeness of the cytoreduction score embodied significant prognostic indicators of survival after HIPEC treatment [34]. Furthermore, Gomes David et al. attempted to determine whether cytoreductive surgery combined with HIPEC is superior to cytoreductive surgery alone in treating endometrial peritoneal carcinomatosis and concluded that, regarding the disease-free survival and overall survival compared to cytoreductive surgery alone in patients with primary or recurrent peritoneal metastases of endometrial cancer, the use of HIPEC combined with cytoreductive surgery had no statistical significance [35]. Similarly, Navarro-Barrios et al. tested the advantages of cytoreductive surgery, alongside HIPEC, in a total of 43 patients with peritoneal metastases and endometrial cancer and reported a recurrence-free survival at 5 years of 23%, with preoperative chemotherapy, more than 3 peritoneal areas removed, cytoreduction of the upper abdominal cavity, paclitaxel-treated HIPEC, and the presence of metastatic lymph nodes all being associated with a worse recurrence-free survival rate [36]. Moreover, Helm et al. combined cytoreductive surgery with cisplatin-based HIPEC for endometrial carcinoma recurrent within the peritoneal cavity and pointed out that, out of the 5 included patients, 2 were living cancer-free after approximately 30 months and 2 were living with the disease at 12 and 36 months, respectively. Unfortunately, 1 patient died at 3 months without evidence of cancer [37]. In addition, Minareci et al. evaluated a total of 32 patients who underwent cytoreductive surgery plus HIPEC retrospectively, 2 of whom had recurrent endometrial cancer and 30 of whom had epithelial ovarian cancer. The researchers concluded that cytoreductive surgery in combination with HIPEC had acceptable severe morbidity and mortality rates, but did not present any specific outcomes concerning exclusively the endometrial cancer subgroup [38]. Peng et al. presented the case of a poorly differentiated grade 3 endometrioid adenocarcinoma. After tumor recurrence, the patient underwent proctectomy with colon-anal anastomosis and cytoreduction surgery with HIPEC including doxorubicin and paclitaxel. Post-operatively, the patient received adjuvant chemotherapy with topotecan, paclitaxel, lipodox, carboplatin, and immunotherapy. Interestingly enough, the immune risk profiles showed Cluster of Differentiation 4 (CD4), CD4/Cluster of Differentiation 8 (CD8) increase after HIPEC and immunotherapy [39]. Interestingly, Rahja et al. used cytoreductive surgery and HIPEC to treat a total of seven patients with peritoneal metastatic endometrial carcinoma, of whom three were primary cases with synchronous peritoneal metastases from the endometrial tumor and four had recurrent disease after the surgical procedure was complete. In all cases, complete cytoreduction was achieved using the same methods and HIPEC chemotherapy, with the exception of one. The survival interval ranged from 5 to 107 months [40]. Santeufemia et al. described the unusual instance of a wound recurrence from endometrial cancer surgically removed 10 years prior that was successfully treated by complete cytoreductive surgery and cisplatin-based HIPEC after responding to megestrol acetate administration. At the 12-month follow-up, the patient was doing well with no evidence of disease recurrence [41]. Last but not least, Yee et al. performed early intraperitoneal chemotherapy on one patient with endometrioid adenocarcinoma. The patient needed to receive one repeat cytoreductive surgery and HIPEC, after which she luckily remained disease-free [42].

Taken altogether, HIPEC seems to exhibit a promising role in the treatment of mostly recurrent endometrial cancer with peritoneal metastasis.

Table 1 briefly summarizes the aforementioned findings.

### 3.2. The Role of HIPEC in Uterine Sarcoma Therapy

A total of 12 articles reported on the role of HIPEC in uterine sarcoma therapy.

Baratti et al. reviewed a database of peritoneal sarcomatosis patients who underwent cytoreductive surgery and close-abdomen HIPEC with cisplatin and doxorubicin or mitomycin-C and concluded that uterine leiomyosarcoma is associated with the higher proportion of long survivors, as well as the best local–regional-free survival [43]. Furthermore, Chetverikov et al. published the case of a 61-year-old woman with relapsed uterine sarcoma who underwent multiple surgical operations, adjuvant chemotherapy, and HIPEC, hence achieving an overall survival rate of 69 months and practically exceeding the theoretically unattainable 5 years from the disease onset [44]. Moreover, Díaz-Montes et al. examined whether patients with recurrent uterine sarcoma who received cytoreductive surgery along with HIPEC had a higher chance of surviving than those who received standard medical care, and found out that both median disease-free and overall survivals were significantly higher for patients treated with HIPEC [45]. Additionally, Düzgün et al. included in their study twenty-two cases of uterine-peritoneal carcinomatosis (mostly uterine sarcoma, but also endometrial cancer) who had undergone cytoreductive surgery and HIPEC, and suggested that, because uterine cancer patients have low peritoneal carcinomatosis index scores and manageable complication rates, cytoreductive surgery and HIPEC should be preferred in peritoneal carcinomatosis due to longer disease-free and overall survival [46]. Inoue et al. performed a debulking surgery and HIPEC in the management of a recurrent aggressive uterine myxoid leiomyosarcoma with peritoneal dissemination and stated that HIPEC with a regimen consisting of the chemotherapeutics cisplatin, VP-16, and mitomycin C, may only be promising in case complete cytoreductive surgery is performed. Importantly, melphalan may be employed as an alternative agent for HIPEC in patients with recurrent peritoneal sarcomatosis, thereby providing meaningful outcomes and survival [47]. Jimenez et al. identified three patients with recurrent high-grade uterine sarcoma (two leiomyosarcomas and one adenosarcoma with sarcomatous overgrowth) with peritoneal dissemination who were treated with cytoreductive surgery and adriamycin/cisplatin- or melphalan-based HIPEC after unsuccessful standard surgical and systemic chemotherapy treatment. After treatment, two of the three patients had no evidence of disease, whereas one patient had disease but was still alive at the end of the follow-up period [48]. The study group of Kasamura et al. assessed the viability and effects of cytoreductive surgery followed by cisplatin/mitomycin-C- or cisplatin/doxorubicin-based HIPEC in ten patients with uterine sarcoma. Overall and progression-free survival rates after 5 years were 65% and 30%, respectively. No operative morbidity, mortality, or toxicity existed. Disease progression was present in six patients [49]. In 2014, Sardi et al. published the promising results of their first study on the use of cytoreduction surgery, alongside melphalan-based HIPEC, for the treatment of peritoneal carcinomatosis, including two cases of uterine sarcoma patients [50]. Four years later, the same study group published the results of two consecutive studies on the role of cytoreductive surgery and HIPEC in women with peritoneal sarcomatosis from uterine sarcoma. More accurately, they suggested that histopathological subtype may influence overall and progression-free survival, with uterine leiomyosarcoma patients exhibiting higher survival rates in comparison with endometrial stromal sarcoma or adenosarcoma patients. In addition, complete surgical tumor excision positively correlated with post-therapy patient survival [51,52]. Spiliotis et al. published two papers on the promising role of HIPEC in recurrent gynecological cancer therapy, including mostly ovarian cancer but also several uterine sarcoma cases [53,54]. In 2016, Sugarbaker et al. published their first original research article on the role of cytoreductive surgery, in combination with HIPEC, for the treatment of patients with disseminated uterine leiomyosarcoma after morcellation or slicing. Of note, early intervention after morcellation correlated with a lesser extent of cancer, while no severe morbidity or mortality was observed in early referral women [55]. Three years later, the same study group reported on the case of a relapse-free uterine leiomyosarcoma with peritoneal metastases, which was treated with surgical resection followed by HIPEC [56]. Yasukawa et al. proposed cytoreductive surgery and HIPEC plus early postoperative intraperitoneal chemotherapy followed by adjuvant systemic chemotherapy as a novel and most promising therapeutic regime for patients with uterine leiomyosarcoma post-morcellation, especially in terms of the 5-year overall survival [57]. Lastly, Zajonz et al. reported a case of a 2-year-old child with a relapse of alveolar rhabdomyosarcoma of the uterus and peritoneal carcinomatosis that was treated with cytoreductive surgery and cisplatin-based HIPEC and died 21 months after treatment due to another rapidly progressive recurrence of the tumor [58].

Altogether, HIPEC seems to play a significant role in the treatment of mostly recurrent uterine leiomyosarcoma with peritoneal sarcomatosis.

Table 2 briefly summarizes the aforementioned findings.

## 4. Discussion and Conclusions

Higher-stage uterine cancers are aggressive gynecologic tumors with a high degree of malignancy and a relatively poor prognosis [1,14]. In distant Surveillance, Epidemiology, and End Results (SEER) stages, the five-year relative survival rates for advanced and/or recurrent uterine cancer are far from satisfactory [1,14]. Importantly, each uterine cancer histologic subtype also shows a unique clinical course, may only be accurately diagnosed postoperatively, and can be challenging to even differentiate from similar benign lesions [59]. As such, uterine cancer still represents a diagnostic and therapeutic challenge that seemingly requires more effective therapeutic approaches, especially in cases of disease progression and/or relapse with peritoneal metastasis. In the present review of the literature, we highlight the role of HIPEC in the treatment of endometrial cancer and uterine sarcoma. To our knowledge, the current work represents the most up-to-date comprehensive literature review on this topic and includes a total of 26 relevant original research articles.

In terms of endometrial cancer, cytoreductive surgery, alongside HIPEC, seems to represent a promising treatment regime for patients with peritoneally metastasized tumor. More precisely, treated patients seem to profit from not only higher survival but also lower recurrence rates. Factors including the completeness of cytoreductive surgery, the peritoneal cancer index, or the applied chemotherapeutic agent, all influence HIPEC therapy effectiveness. In the context of uterine sarcoma, cytoreductive surgery in combination with HIPEC also seems to represent a revolutionizing novel treatment concept for tumors with peritoneal sarcomatosis, with the histologic subtype, the chosen chemotherapeutic agent, or the percentage of the cytoreduction, significantly determining the post-treatment patient survival. Most importantly, HIPEC is not associated with increased operative morbidity, mortality, or toxicity, hence constituting a safe novel therapeutic alternative for selected uterine cancer patients.

Even though HIPEC has, to date, been mainly employed for the management of advanced ovarian cancer, the present review of the literature outlines its advantages for the therapy of other gynecologic malignancies with a special focus on advanced uterine cancer. Nevertheless, given that current studies incorporate only small patient samples, future clinical trials should incorporate larger patient collectives in order to confirm the reported first preliminary outcomes and to also reliably investigate multiple factors co-determining the therapeutic outcome. Importantly, eventual side effects of HIPEC, alongside comorbidities of the study population, need to be taken into consideration in terms of treatment failure assessment.

The nonsystematic methodology used in the context of study selection is one of the review’s limitations. Although rigorous rules and the standards-compliant systematic literature reviews offer the most accurate method for identifying pertinent research works, this approach necessitates a specific research question that excludes broader subjects like the role of HIPEC in uterine cancer therapy. The eventual evidence selection bias, which results from publication bias because data from statistically significant studies are more likely to be published, is another restriction. Additionally, only one person and two databases were used for the literature analysis. Last but not least, original research articles that might be interesting or pertinent but were not written in English had to be disregarded.

In conclusion, HIPEC could be regarded as a groundbreaking treatment alternative for patients with advanced uterine cancer, able to ameliorate overall and relapse/progression-free survival of the affected tumor patients.

## Figures and Tables

**Figure 1 ijms-24-12353-f001:**
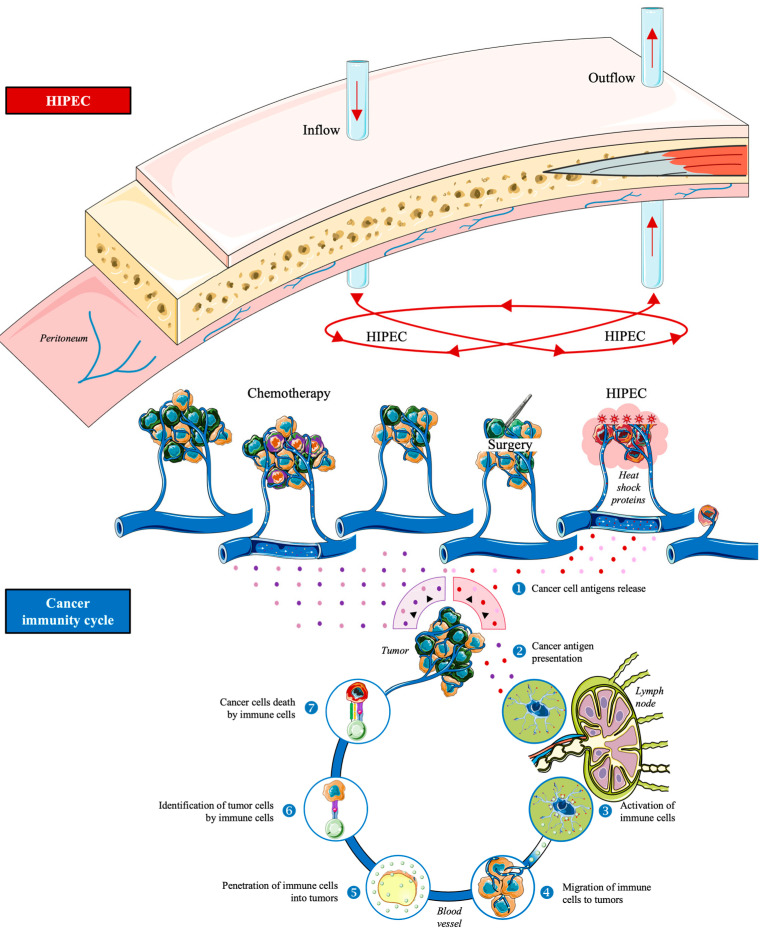
Mechanism of action of HIPEC in peritoneal carcinomatosis.

**Figure 2 ijms-24-12353-f002:**
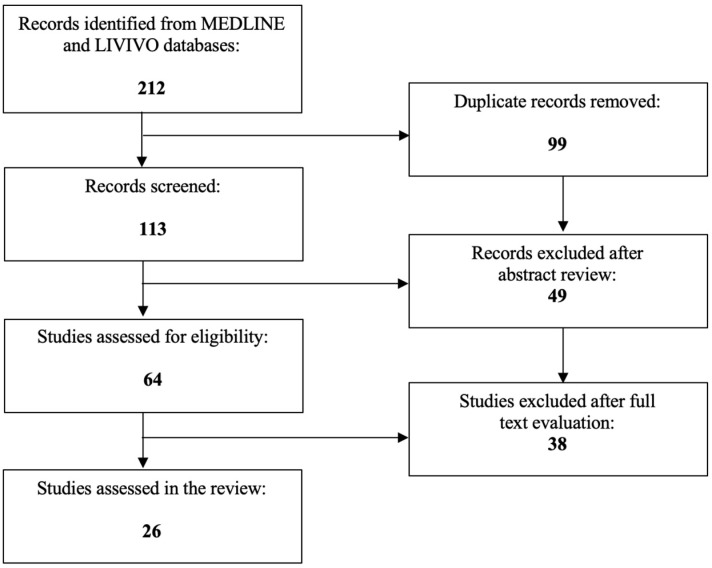
PRISMA flow diagram visually summarizing the screening process.

**Table 1 ijms-24-12353-t001:** The role of HIPEC in endometrial cancer therapy.

Study	Patient Collective	Post-Treatment Outcomes
Abu-Zaid et al. [29]	6 patients with peritoneal carcinomatosis arising from endometrial cancer	1/6 experienced disease relapse with metastases and died 1/6 hepatic metastases within three months, but was still alive at follow-up 4/6 were alive and disease free without evidence of recurrence
Bakrin et al. [30]	5 patients with recurrent endometrial cancer	1/5 experienced recurrent disease 10 months post-operatively and died two months later 1/5 developed early recurrence with a malignant pleural effusion and died 3/5 patients were alive and disease free
Brind’Amour et al. [31]	3 patients with endometrial cancer and synchronous peritoneal metastases	2/3 died 1/3 did not show a disease relapse
Chambers et al. [32]	7 uterine serous carcinoma patients	Good treatment tolerance Median progression-free survival of 14 months Median overall survival of 27 months
Cornali et al. [33]	33 patients with peritoneal metastases from endometrial cancer	Completeness of cytoreduction with a low peritoneal cancer index as the only significant independent factor determining overall survival 5-year overall survival of 30% Progression-free survival of 15.5%
Delotte et al. [34]	13 endometrial cancer patients	3/13 died within the first year of treatment 2/13 died 12 and 19 months after their HIPEC procedure Between one and 125 months after surgery, 7/13 patients were still alive, 4 of whom had not experienced a recurrence Peritoneal cancer index and the completeness of cytoreduction score are significant prognostic indicators of survival after HIPEC treatment
Gomes David et al. [35]	74 patients with peritoneal metastases of endometrial cancer	Compared to cytoreductive surgery alone in patients with primary or recurrent peritoneal metastases of endometrial cancer, the use of HIPEC combined with cytoreductive surgery had no statistical significance
Navarro-Barrios et al. [36]	43 patients with peritoneal metastases and endometrial cancer	Recurrence-free survival at 5 years of 23% Preoperative chemotherapy, more than three peritoneal areas removed, cytoreduction of the upper abdominal cavity, paclitaxel-treated HIPEC, and the presence of metastatic lymph nodes corelate with a worse recurrence-free survival rate
Helm et al. [37]	5 patients with endometrial carcinoma recurrent within the peritoneal cavity	2/5 cancer free after approximately 30 months 2/5 living with disease at 12 and 36 months, respectively 1/5 died at 3 months without evidence of cancer
Minareci et al. [38]	2 patients with recurrent endometrial cancer	Acceptable severe morbidity and mortality rates
Peng et al. [39]	Poorly differentiated grade 3 endometrioid adenocarcinoma patient	CD4, CD4/CD8 increase after HIPEC and immunotherapy
Rahja et al. [40]	7 patients with peritoneal metastatic endometrial carcinoma	Survival interval range from 5 to 107 months
Santeufemia et al. [41]	Wound recurrence from a surgically removed endometrial cancer	No evidence of disease recurrence
Yee et al. [42]	1 patient with endometrioid adenocarcinoma	Disease-freedom after repeat cytoreductive surgery and HIPEC

**Table 2 ijms-24-12353-t002:** The role of HIPEC in uterine sarcoma therapy.

Study	Patient Collective	Post-Treatment Outcomes
Baratti et al. [43]	11 patients with uterine leiomyosarcoma	Uterine leiomyosarcoma is associated with the higher proportion of long survivors, as well as the best local–regional-free survival
Chetverikov et al. [44]	1 patient with relapsed uterine sarcoma	Overall survival rate of 69 months
Díaz-Montes et al. [45]	26 patients with recurrent uterine sarcoma	Both median disease-free and overall survivals are significantly higher for patients treated with HIPEC
Düzgün et al. [46]	22 cases of uterine-peritoneal carcinomatosis	HIPEC should be preferred in peritoneal carcinomatosis due to longer disease-free and overall survival
Inoue et al. [47]	1 recurrent aggressive uterine myxoid leiomyosarcoma with peritoneal dissemination	Complete cytoreductive surgery is required Melphalan may be employed as an alternative agent for HIPEC in patients with recurrent peritoneal sarcomatosis
Jimenez et al. [48]	3 patients with recurrent high-grade uterine sarcoma	2/3 had no evidence of disease 1/3 had disease, but was still alive
Kasamura et al. [49]	10 patients with uterine sarcoma	Overall and progression-free survival rates after five years were 65% and 30%, respectively No operative morbidity, mortality, or toxicity existed. Disease progression was present in 6/10
Sardi et al. [50,51,52]	45 patients with peritoneal sarcomatosis from uterine sarcoma	Melphalan as a promising HIPEC agent Histopathological subtype may influence overall and progression-free survival Uterine leiomyosarcoma patients exhibit higher survival rates in comparison with endometrial stromal sarcoma or adenosarcoma patients Complete surgical tumor excision positively correlates with post-therapy patient survival
Spiliotis et al. [53,54]	3 uterine sarcoma cases	Acceptable mortality and morbidity Improved survival
Sugarbaker et al. [55,56]	8 uterine leiomyosarcoma patients	Early intervention after morcellation correlated with a lesser extent of cancer No severe morbidity or mortality
Yasukawa et al. [57]	6 uterine leiomyosarcoma patients	Cytoreductive surgery and HIPEC plus early postoperative intraperitoneal chemotherapy followed by adjuvant systemic chemotherapy as a novel and most promising therapeutic regime for patients with uterine leiomyosarcoma post-morcellation
Zajonz et al. [58]	Relapse of an alveolar rhabdomyosarcoma of the uterus and peritoneal carcinomatosis in a 2-year-old child	Death 21 months after treatment due to another rapidly progressive recurrence of the tumor

## Data Availability

Not applicable.

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
