# Peer review of "The Role of Hyperthermic Intraperitoneal Chemotherapy in Uterine Cancer Therapy"

_ijms, 2023, doi:10.3390/ijms241512353_

Round 1

Reviewer 1 Report

innovative approach being described and evidence-based as it is recommended in severe dg. It is important that all aspects of the treatment are clearly described with all the risks and benefits (side effects) which the authors should have been putting more emphasis on. see comments inside the manuscript for minor revisions.

Author Response

Innovative approach being described and evidence-based as it is recommended in severe dg. It is important that all aspects of the treatment are clearly described with all the risks and benefits (side effects) which the authors should have been putting more emphasis on. see comments inside the manuscript for minor revisions.

Thank you for your comments and suggestions.

  1. We have improved the mentioned type of the review, it is actually a narrative and not a systematic review.
  2. The fact that the full texts of the remaining 64 publications were assessed and that a total of 26 relevant studies meeting the inclusion criteria and published between 2004 and 2023 were selected for the final literature review, is mentioned in the abstract and again in the main text as part of the methods.
  3. We have now split the long sentence in two shorter sentences.
  4. We have now added a sentence in the discussion, highlighting the named factors that contribute to treatment failure.
  5. In the last paragraph of the discussion we highlight our nonsystematic approach.

Reviewer 2 Report

1. The manuscript reviewed the effects of HIPEC on uterine cancer therapy, and the outcomes of HIPEC on patients with endometrial cancer and sarcoma were listed, and could be usful for readers.

2. Total reviewed articles counted only for 26 reports, the authors might could consider to broaden the screening conditions and/or eligibility standards that increase the credibility. 

3. The molecular mechanisms of endometrial cancer and sarcoma were inclided in Introdiction; however, the influenced pathways that cause side-effects and/or metastasis during chemotherapy were not mentioned. The informations might cause research inspirations for further usages of HIPEC.

Author Response

1. The manuscript reviewed the effects of HIPEC on uterine cancer therapy, and the outcomes of HIPEC on patients with endometrial cancer and sarcoma were listed, and could be useful for readers.

Thank you for your comment.

2. Total reviewed articles counted only for 26 reports, the authors might could consider to broaden the screening conditions and/or eligibility standards that increase the credibility. 

Indeed, the total reviewed articles counted only for 26 reports. To our knowledge, the current work represents, however, the most up-to-date comprehensive literature review on this specific topic. The chosen eligibility standards allow for a high-quality review of the literature, hence minimizing common limitations that reduce the reliability of the study results.

3. The molecular mechanisms of endometrial cancer and sarcoma were included in Introduction; however, the influenced pathways that cause side-effects and/or metastasis during chemotherapy were not mentioned. The information might cause research inspirations for further usages of HIPEC.

We have now added the influenced pathways during HIPEC in the introduction.

Round 2

Reviewer 2 Report

The manuscript was revised and provide useful information.